


# Measurement Report: Spectral and statistical analysis of aerosol hygroscopic growth from multi-wavelength lidar measurements in Barcelona, Spain

**Michaël Sicard[1,2], Daniel Camilo Fortunato dos Santos Oliveira[1], Constantino Muñoz-Porcar[1], Cristina Gil-Díaz[1], Adolfo Comerón[1], and Alejandro Rodríguez-Gómez[1], Federico Dios Otín[1]**

[1]CommSensLab, Dept. of Signal Theory and Communications, Universitat Politècnica de Catalunya (UPC), 08034-Barcelona, Spain
[2]Ciències i Tecnologies de l'Espai-Centre de Recerca de l'Aeronàutica i de l'Espai/Institut d'Estudis Espacials de Catalunya
(CTE-CRAE/IEEC), Universitat Politècnica de Catalunya (UPC), 08034-Barcelona, Spain

*Correspondence to*: Michaël Sicard (michael.sicard@upc.edu)

**Abstract.** This paper presents the estimation of the hygroscopic growth parameter of atmospheric aerosols retrieved with a multi-wavelength lidar, a micro pulse lidar and daily radiosoundings in the coastal region of Barcelona, Spain. The hygroscopic growth parameter, $\gamma$, parametrizes the magnitude of the scattering enhancement in terms of the backscatter coefficient

following Hänel parametrization. After searching for time co-located lidar and radiosoundings measurements, a strict criterion-based procedure (limiting the variations of magnitudes such as water vapor mixing ratio, potential temperature, wind speed and direction) is applied to select only cases of aerosol hygroscopic growth. A spectral analysis (at the wavelengths of 355, 532 and 1064 nm) is performed with the multi-wavelength lidar, and a climatological one, at the wavelength of 532 nm, with the database of both lidars. The spectral analysis shows that below 2 km (regime of local pollution and sea salt) $\gamma$ decreases

with increasing wavelengths. This behaviour can be attributed to the aerosol size: the smaller the aerosol, the more hygroscopic. Above 2 km (regime of regional pollution and residual sea salt) the values of $\gamma$ at 532 nm are nearly the same than below 2 km, and its spectral behaviour is flat. This analysis and others from the literature are put together in a table presenting, for the first time, a spectral analysis of the hygroscopic growth parameter of a large variety of atmospheric aerosol hygroscopicities going from low (pure mineral dust, $\gamma < 0.2$) to high (pure sea salt, $\gamma > 1.0$) hygroscopicity. The climatological analysis shows

that, at 532 nm, $\gamma$ is rather constant all year round and has a large monthly standard deviation suggesting the presence of aerosols with different hygroscopic properties all year round. The annual $\gamma$ is 0.55±0.23. The height of the hygroscopic layers shows an annual cycle with a maximum in summer and a minimum in winter. Former works describing the presence of re-circulation layers of pollutants injected at various heights above the PBL may explain why $\gamma$, unlike the height of the hygroscopic layers, is not season-dependent. The sub-categorization of the whole database into *No cloud* and *Below-cloud*

cases reveals a large difference of $\gamma$ in autumn between both categories (0.71 and 0.33, respectively), possibly attributed to a



depletion of inorganics at the point of activation into cloud condensation nuclei in the *Below-cloud* cases. Our work calls for more in-situ measurements to synergetically complete such studies based on remote sensing.

## 1 Introduction

Atmospheric aerosols and water vapor are atmospheric components of extreme importance for the climate on Earth.
Atmospheric aerosols influence the energy balance between the Earth and the atmosphere, directly through their scattering and absorbing interaction with the electromagnetic radiation (Thorsen et al., 2020), and indirectly modifying the thermodynamic profiles (semi-direct effect, Hansen et al., 1997; Koren et al., 2004) or changing cloud properties, including their lifetime and albedo (indirect effects, Seinfeld et al., 2016). Generally, the aerosol effects on the Earth-atmosphere energy budget depend on the aerosol optical, microphysical and radiative properties, their life time in the atmosphere, synoptic conditions and other
factors like water vapor. The latter, the most important primary component for the formation of clouds to occur, has also an effect on the aerosol size distribution and thus on their optical and microphysical properties. Indeed some aerosol types may increase in size due to water uptake under high relative humidity (RH) conditions. This process is called hygroscopic growth (Hänel, 1976) and it is determined mainly by the aerosol chemical composition and in particular by the mixing of inorganic and organic components (Sjogren et al., 2007). Remarkably, the hygroscopic growth plays an important role in the aerosol-
cloud interaction (Kanakidou et al., 2005).

The capability of aerosols to grow hygroscopically is linked to their chemical composition. Atmospheric aerosols can be classified as hydrophobic (e.g. dust) or hydrophilic with monotonic (smoothly varying, e.g. volcanic) or deliquescent (step change, e.g. marine) growth (Carrico et al., 2003). For deliquescent cases, a hygroscopic dry material exposed to increasing RH will start to grow in size only when the deliquescence RH is reached. The deliquescence RH corresponds to the equilibrium
RH over an aqueous saturated solution with respect to its solute. Further increases in RH result in continued droplet growth. After deliquescence and upon exposure to decreasing RH, the aqueous droplet can form a metastable droplet, supersaturated with respect to solute concentration, until a lower crystallization (also called efflorescence by other authors, e.g. Sjogren et al., 2007) RH is reached. As a result, the humidogram (the representation of a variable as a function of RH) of the particle size or of its optical properties usually presents a strong hysteresis with a lower branch (humidification) and upper branch
(dehydration), that intercept at or near the deliquescent and crystallization RH. As an example, measurements of pure salts of NaCl performed in the field during ACE-Asia yielded a deliquescent and crystallization RH of 75 and 41 % (Carrico et al., 2003), respectively.

The response of aerosols to changes in RH can be measured by a variety of instruments. In the last decade active remote sensing systems, e.g. like lidars, have revealed to be an adequate technique for the identification and analysis of the aerosol
hygroscopic growth compared to more traditional instruments like, e.g., humidified nephelometers and spectrometers. The advantages of active remote sensing systems are multiple: they provide high vertical and temporal resolutions, they preserve the ambient conditions (no humidification or dehydration is applied on the sample), they can measure under relative humidities



close to saturation. In the literature the aerosol hygroscopic enhancement has been measured more often on the backscatter coefficient derived from lidar (Feingold and Morley, 2003; Fernández et al., 2015; Granados-Muñoz et al., 2015; Haarig et al.,

2017; Lv et al., 2017; Navas-Guzmán et al., 2019; Chen et al., 2019; Pérez-Ramírez et al., 2021) and ceilometer (Bedoya-Velásquez et al., 2019) measurements than on the extinction coefficient derived from lidar measurements (Veselovskii et al., 2009; Dawson et al., 2020). Some intents to work on the attenuated backscatter coefficient derived from ceilometers were performed by Haeffelin et al. (2016) to help tracking the activation of aerosols into fog or low-cloud droplets. Others investigated the lidar ratio changes due to relative humidity and their effect on the classical elastic-backscatter lidar inversion

technique (Zhao et al., 2017).

The present work takes advantage of observational capabilities quite unique at the site of Barcelona, NE Spain: a multi-wavelength lidar system measuring at three elastic wavelengths since 2011 and a single wavelength micro pulse lidar working continuously 24/7 since 2015, as well as two radiosoundings launched everyday almost collocated to the lidars (the database starts in 2009). The paper deals with 1) the spectral analysis of the hygroscopic growth factor measured at three wavelengths,

and 2) the climatological analysis of the hygroscopic growth measured at 532 nm in Barcelona. The spectral analysis is motivated by conclusions from Dawson et al. (2020) who say that multispectral lidars are fundamental so as to "provide additional insight into the [hygroscopic enhancement factor] retrievals since the 355-nm wavelength is sensitive to smaller aerosols than the 532-nm wavelength". The climatological analysis is a partial answer to the call of several authors, e.g. like Bedoya-Velásquez et al. (2018), for further investigation extending the study periods to obtain results statistically more robust.

The structure of the paper is as follows: Section 2 describes the instrumentations and the methodology and Section 3 presents the results of the spectral and climatological analysis. Conclusions are given in Section 4.

## 2 Instrumentation and methodology

### 2.1 Lidars and radiosoundings in Barcelona

All measurements presented in this paper were performed at or close to the Barcelona lidar site at the Remote Sensing

Laboratory of the Department of Signal Theory and Communications at the Universitat Politècnica de Catalunya (41.393ºN, 2.120ºE, 115 m asl). Two lidar systems were used: the multi-wavelength ($3\beta+2\alpha+2\delta+WV$) ACTRIS/EARLINET lidar and the micro pulse lidar (MPL, $1\beta+1\delta$). The first system is run according to a regular weekly schedule and to monitor special aerosol events of interest. Aerosol optical properties from this system can be found in the ACTRIS database at https://actris.nilu.no/. The system employs a Nd:YAG laser emitting pulses at 355, 532 and 1064 nm at a repetition frequency of 20 Hz. The

measurements of the ACTRIS/EARLINET system are averaged over 30 or 60 minutes. The retrieved backscatter coefficients at the three emitted wavelengths for the period 2010-2018 are used in this work. General details about the system can be found in Kumar et al. (2011). The MPL system runs continuously 24/7. It is part of the Micro-Pulse Lidar Network (MPLNET, https://mplnet.gsfc.nasa.gov/data?v=V3; Welton et al., 2001) since 2016. The system uses a pulsed solid-state laser emitting low-energy pulses (~6 μJ) at a high pulse rate (2500 Hz). All MPL measurements are averaged over 60 minutes. The MPL



data used in this work are the backscatter coefficient profiles at 532 nm retrieved during the period 2015-2018. More technical
information about the system can be found in Campbell et al. (2002), Flynn et al. (2007) and Welton et al. (2018). All the MPL
retrievals presented in this work were performed with in-house algorithms, and not with the MPLNET processing.

Radiosoundings measurements are launched twice a day (at 00:00 and 12:00 UTC) by the the Meteorological Service of
Catalonia, Meteocat, at a distance of less than 1 km from the lidar site. The radiosoundings provide measurements of pressure,

temperature, relative humidity and wind speed and direction. Data of the period 2010-2018 are used in the present work. At
this point, it is important to note the inherent spatial drift of radiosoundings and the long integration time of the lidar data (as
long as 60 minutes) which may cause a loss of temporal and spatial coincidence between both retrievals. This effect can be
enhanced during daytime when the atmosphere may change quickly. This has been demonstrated by a recent paper from
Muñoz-Porcar et al. (2021) in which profiles of water vapor mixing ratio retrieved with lidar and radiosoundings were

compared. The authors also highlighted the high variability of the profile of relative humidity in Barcelona due to the presence
of the sea coast, the mild temperatures of the Mediterranean climate inducing regularly land-to-sea and sea-to-land breeze
regimes and the local orography.

## 2.2 Methodology

This paper deals with the enhancement factor of the particle backscatter coefficient, $\beta$, as a function of relative humidity, $RH$,

commonly noted $f_\beta(RH)$ in the literature. Since no other optical/microphysical property is considered here, the $\beta$ suffix is
omitted in the rest of the paper in order to alleviate the formulae. The wavelength dependency is indicated with a superscript
$\lambda$. Finally, the backscatter coefficient at wavelength $\lambda$ writes $\beta^\lambda$ and the corresponding enhancement factor writes $f^\lambda(RH)$.

The methodology starts with the search of time co-located lidar and radiosoundings measurements within a difference smaller
than ±120 minutes. The time co-located measurements are then analysed to look for vertical intervals $(h_{min}, h_{max})$ in which

a monotonic increase of the particle backscatter coefficient and of the relative humidity simultaneously occurs. After fulfilling
the initial conditions, these vertical intervals are classified as hygroscopic growth cases following a strict criterion-based
procedure including:

- water vapor mixing ratio noted WVMR (maximum variation of 2 g kg⁻¹),
- potential temperature noted $\theta$ (maximum variation of 2 K),

- wind speed (maximum variation of 2 m s⁻¹),
- wind direction (maximum variation of 15 deg.).

For all cases back trajectories were also calculated with HYbrid Single-Particle Lagrangian Integrated Trajectory (HYSPLIT)
model (Stein et al., 2015) in order to verify the aerosol origin inside the layers of the selected cases obtained by applying the
criterion-based procedure. These criteria guarantee that the increase of $RH$ is very likely due to an increase of the water vapor

and not to a combined effect of the thermodynamics variables, and that the variations of $\beta^\lambda$ are caused by an increase of the





aerosol size due to water uptake and not to changes in the aerosol composition or concentration in the analysed layer. Such criteria have been applied by other authors (Granados-Muñoz et al., 2015; Navas-Guzmán et al., 2019; among others).

In order to discard the cases including mineral dust which is known to be poorly hydrophilic, in case of doubts, the back trajectory analysis was completed with mineral dust forecast from the NMMB/BSC-Dust model (https://ess.bsc.es/bsc-dust-

daily-forecast) and AERONET retrievals.

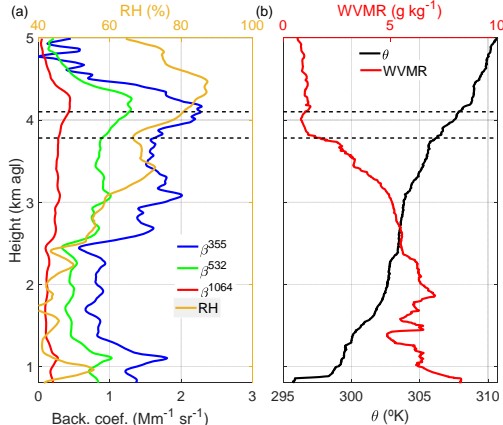

**Figure 1** Vertical profiles of (a) $\beta_\lambda$ at 3 wavelengths and $RH$; (b) WVMR and $\theta$. The horizontal dash lines indicate $h_{min}$ and $h_{max}$ obtained by applying the criterion-based procedure. The example is from 22 July, 2013, at 13:02 UTC.

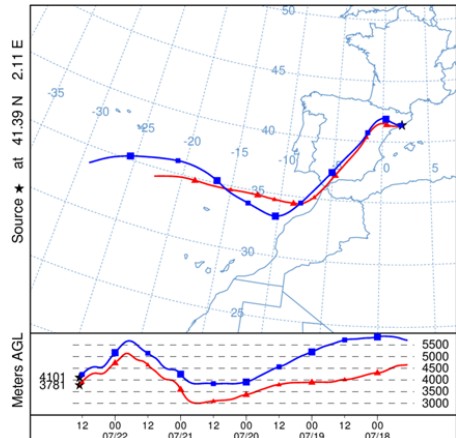


**Figure 2** 5-day back trajectories arriving in Barcelona on 22 July, 2013, at 13:00 UTC, at $h_{min}$ = 3781 m and $h_{max}$ = 4101 m agl.





Figure 1 is an example of one selected hygroscopic growth case. It shows the vertical profiles of $\beta^\lambda$ at 3 wavelengths, $RH$, WVMR and $\theta$. A simultaneous increase of $\beta^\lambda$ and $RH$ is observed inside the layer between $h_{min}$= 3781 m and $h_{max}$= 4101

m. The maximum variations of WVMR and $\theta$ inside this layer are 0.91 g kg⁻¹ and 1.63 K, respectively. The back trajectories arriving in Barcelona at $h_{min}$ and $h_{max}$ and shown in Figure 2 demonstrate that the air masses at the base and top of the considered layer come from same origins, Atlantic Ocean and continental zones, ensuring that the aerosols inside the whole layer come from the same sources.

For each case, each value of $\beta^\lambda$ in the range $[h_{min}, h_{max}]$ has a corresponding value of $RH$ varying in the range $[RH_{min},$

$RH_{max}]$. For each case, we define the particle backscatter coefficient enhancement factor, $f_{min}^\lambda(RH)$, defined starting from $RH_{min}$ as:

$$f_{min}^\lambda(RH) = \frac{\beta^\lambda(RH)}{\beta^\lambda(RH_{min})} \tag{1}$$

$f_{min}^\lambda(RH)$ quantifies the increase of $\beta^\lambda$ when the relative humidity increases from $RH_{min}$ to $RH$. A fitting of $f_{min}^\lambda(RH)$ is performed with the so-called Hänel parametrization (Kasten, 1969; Sheridan et al., 2002) using the points available between

$RH_{min}$ and $RH_{max}$:

$$f_{min}^\lambda(RH) = \left(\frac{1 - RH/100}{1 - RH_{min}/100}\right)^{-\gamma(\lambda)} \tag{2}$$

where $\gamma(\lambda)$ is the hygroscopic growth parameter and parameterizes the magnitude of the scattering enhancement. This definition of the enhancement factor limits the range of relative humidity values for which it can be calculated, i.e. only for $RH > RH_{min}$, which prevents direct comparisons when $RH_{min}$ is not the same (Veselovskii et al., 2009). In order to extend

this range and have results comparable in the same range of relative humidity values, the different functions of $f_{min}^\lambda(RH)$ are scaled to a larger range of values of $RH$: [40, 90 %]. The bottom limit, 40 %, is noted $RH_{ref}$. This reference value of 40 % has been used in several works (Skupin et al., 2016; Titos et al., 2016; Haarig et al., 2017; Bedoya-Velásquez et al., 2018; Dawson et al., 2020). $RH_{ref}$=40 % is a recommendation of World Meteorological Organization (2016) who demonstrates with in-situ measurements that the hygroscopicity growth effect on aerosols is minimized for values of the relative humidity below

40 %. The new scaled enhancement factor starting at $RH_{ref}$=40 % is noted $f_{ref}^\lambda(RH)$ and expresses the increase of $\beta^\lambda$ when the relative humidity increases from $RH_{ref}$ to $RH$:

$$f_{ref}^\lambda(RH) = \left(\frac{1 - RH/100}{1 - RH_{ref}/100}\right)^{-\gamma(\lambda)} \tag{3}$$

However, when $RH_{min} \neq RH_{ref}$ this function cannot be calculated directly. A solution is to calculate it from $f_{min}^\lambda(RH)$ as follows:


$$f_{ref}^{\lambda}(RH) = f_{min}^{\lambda}(RH)\left(\frac{1 - RH_{min}/100}{1 - RH_{ref}/100}\right)^{-\gamma(\lambda)} \qquad (4)$$


Note that the term on the right-hand side multiplying $f_{min}^{\lambda}(RH)$ is nothing else than the quotient of the intercepts of both enhancement factors ($f_{min}^{\lambda}$ and $f_{ref}^{\lambda}$) at $RH=0$ %. It is the first time a conversion of $f_{min}^{\lambda}$ into $f_{ref}^{\lambda}$ is proposed. Figure 3 is an example showing the two retrieved enhancement factors $f_{min}^{\lambda}$ and $f_{ref}^{\lambda}$ and their Hänel fit. One sees clearly the difference between $f_{min}^{\lambda}$ restricted to $[RH_{min}, RH_{max}]$ (spanning 21 % in the example presented) and $f_{ref}^{\lambda}$ spanning 50 % from 40 to

90%. In addition, to allow direct comparison of enhancement factors retrieved from different cases for different $RH$ ranges, this method has also the advantage of defining a common way for the calculation of the $f$-value. The $f$-value, also called the $f(RH)$ value (Titos et al., 2016), is defined as $f_{min}^{\lambda}(RH_{max})$. It depends on both $RH_{min}$ and $RH_{max}$. There is no consensus in the atmospheric community for the definition of the range of $RH$ values which has a strong variability among studies as underlined by Titos et al. (2016). In this study, the $f$-value is $f_{ref}^{\lambda}(RH = 85\%)$ with $RH_{ref}$=40 %, and it applies for all cases.

It expresses the increase factor of the backscatter coefficient when the relative humidity increases from $RH_{ref}$=40 % to 85 %.

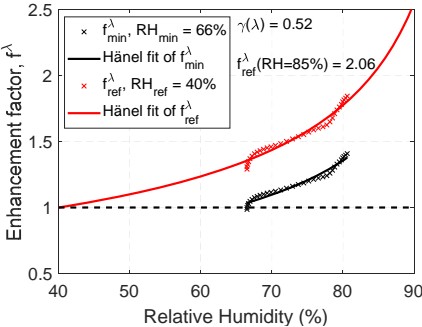

**Figure 3** Example of enhancement factors $\boldsymbol{f_{min}^{\lambda}}$ defined starting at $\boldsymbol{RH_{min}}$ and $\boldsymbol{f_{ref}^{\lambda}}$ defined starting at $\boldsymbol{RH_{ref}}$ = 40 % of one single case (22 July, 2013, at 13:02 UTC; λ=532 nm).

Finally, to avoid outliers the cases with $f_{ref}^{\lambda}(RH = 85\%)$ greater than 10 with $RH_{ref}$=40 % (which corresponds to $\gamma \approx 1.66$)

were not taken into account in the statistics.

### 3 Results and discussion

#### 3.1 Spectral analysis

In this section, only the ACTRIS/EARLINET lidar system, which has three elastic wavelengths, is considered. Among the backscatter profiles at the three wavelengths of 355, 532 and 1064 nm and the relative humidity profiles from radiosoundings

available between 2010 and 2018, 32 potential cases of hygroscopic growth which fulfilled the selection criteria mentioned in





Section 2.2 were identified. From these 32 hygroscopic growth cases, the centre of the layer considered was below 2 km agl in 8 cases (25 %) and above 2 km agl in 24 cases (75 %). This reference height of 2 km was chosen based on Sicard et al. (2006, 2011) and (Pandolfi et al., 2013) who showed that the separation between the surface mixed layers and possible decoupled residual/aloft layers in the Barcelona coastal area are more likely to occur around 2 km high. Interestingly Pérez-

Ramírez et al. (2021) also considered altitude heights below 2 km (near surface and PBL) and above 2 km to show the temporal evolution of $f(RH)$ and $\gamma$. Generally speaking, in the region of Barcelona, the aerosols present below 2 km, i.e. in the PBL, are representative of a coastal, urban background site and their chemical composition is dominated by anthropogenic, crustal and marine aerosols (Querol et al., 2001). Below 2 km, the aerosol type in Barcelona is defined as local pollution and marine aerosols. According to Pey et al. (2010), the mean annual urban contribution in Barcelona downtown of hydrophilic species

such as $SO_4^{2-}$, $NO_3^-$ or $NH_4^+$ is at least 53, 65 and 45 %, respectively, with respect to the regional background. For aerosol layers above 2 km, Sicard et al. (2011) refer to "recirculation polluted air-masses" and Pandolfi et al. (2013) to either regional or Atlantic air-masses (African air-masses are discarded since the cases with the presence of mineral dust are not included in this study). Above 2 km, the aerosol type in Barcelona is defined as regional pollution and marine aerosols. It is important to mention that above 2 km, the height range reaches 5 km, the approximate height where the highest top of the layer is located.

To check the provenance of the aerosol layers below and above 2 km, we plot in Figure 4 the wind roses for the 8 cases below 2 km and the 24 cases above 2 km. While the wind directions below 2 km (Figure 4a) are diverse (NE, SSW, WSW, WNW), they are more clustered above 2 km (Figure 4b) and fall almost all between SW and NNW, indicating clearly a peninsular origin.

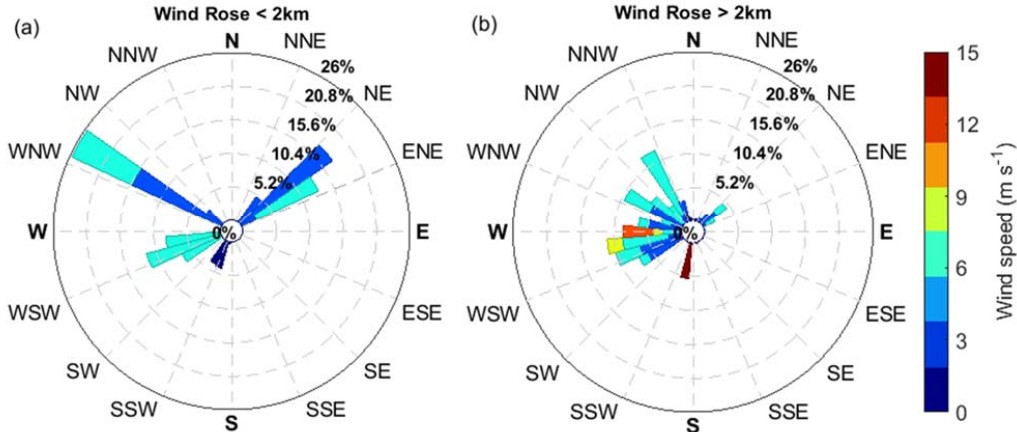

**Figure 4** Wind rose (a) below 2 km (8 cases) and (b) above 2 km (24 cases) from radiosoundings measurements in Barcelona. The colorbar on the right applies to both plots.





For both wind roses, the most frequent wind speeds reach around 6 m s$^{-1}$ designated as moderate breeze (World Meteorological Organization, 1992). This low wind speed maintains a good homogeneity of the atmospheric aerosols in the selected layers, i.e., a constant mixture of aerosols over time.

Figure 5a and 5e show the resulting spectral Hänel fits at both height levels. Figure 5b-d and 5f-h show all enhancement factors per wavelength below and above 2 km agl, respectively. In all those plots, the Hänel fits (solid lines) are calculated with the mean hygroscopic growth parameter $\bar{\gamma}(\lambda)$ of all individuals $\gamma(\lambda)$, and the variability associated to them (coloured shaded area) is calculated taking into account the standard deviation of all individual $\gamma(\lambda)$. The enhancement factors and Hänel fits in Figure 5 are those scaled in order to start at $RH_{ref}$=40 %. The spectral values of $f_{ref}^{\lambda}(RH = 85\%)$ and $\bar{\gamma}(\lambda)$ at both height levels are

reported in Table 1, which also includes the mean of the correlation coefficients, $R^2$, of the individual pairs of $(\beta^{\lambda}, RH)$, as well as the average of the layer-mean Ångström exponents, $AE_{\lambda_1,\lambda_2}$, between the pairs of wavelengths (355, 532 nm) and (532, 1064 nm). The first result to comment is that, independently of the height level, the correlation coefficients of the individual fits are high (>0.91) and present small fluctuation ($\sigma$<0.06, except for $\lambda$=355 nm and above 2 km where $\sigma$=0.14). These high values of $R^2$ indicate the good correlation that exists between the profiles of the backscatter coefficients and the profiles of the

relative humidity in the layer selected.

Below 2 km, the particle backscatter coefficient enhancement factor seems to have a clear spectral behaviour: $f_{ref}^{\lambda}(RH = 85\%)$ $(\gamma(\lambda))$ is 3.60 ± 2.47 (0.81 ± 0.41), 3.18 ± 2.07 (0.73 ± 0.40) and 2.68 ± 1.20 (0.65 ± 0.31) at 355, 532 and 1064 nm, respectively. This behaviour (decrease of $f_{ref}^{\lambda}(RH = 85\%)$ or $\gamma(\lambda)$ with increasing wavelength) implies that the water uptake by the particles modifies the particle backscatter coefficient more strongly at shorter wavelengths than at larger

wavelengths. Since the 355-nm wavelength is sensitive to smaller aerosols compared to larger wavelengths, larger fit coefficients at 355 nm indicate slightly more hygroscopic aerosols present at smaller size ranges (Dawson et al., 2020). $AE_{355,532}$ and $AE_{532,1064}$ are 1.07 and 0.93, respectively. Although quite similar, the difference between both $AE_{\lambda_1,\lambda_2}$ also points out to a spectral sensitivity of the backscatter coefficient slightly larger at shorter wavelengths than at larger ones. Note en passant that our Ångström exponent values are in the range of column-averaged monthly values found by Sicard et al.

(2011) and estimated from a long-term lidar database in Barcelona. Next, we aim at comparing our results with the literature. The hygroscopic growth parameter $\gamma$ depends on neither $RH_{min}$, nor $RH_{max}$, so the values of $\gamma$ from the literature can be directly compared to ours. Contrarily, the $f$-values depend strongly on $RH_{min}$ and $RH_{max}$, so for the literature to be comparable with our values, the hygroscopic growth parameter $\gamma$ from the literature is used to calculate $f_{ref}^{\lambda}(RH = 85\%)$ with $RH_{ref}$=40 %. Also, we only considered works in which the enhancement factor was calculated for the backscatter

coefficient measured with a lidar. Works in which the enhancement factor was calculated for the extinction coefficient (Veselovskii et al., 2009; Dawson et al., 2020) or from in-situ data (Carrico et al., 2003; Titos et al., 2016; Skupin et al., 2016; among others) are not considered for comparison with our study. The literature results are summarized in Table 2 and represented in Figure 6, in terms of $f_{ref}^{\lambda}(RH = 85\%)$ and $\gamma(\lambda)$. The values in Table 2 are ordered from high to low values of





$f_{ref}^{532}(RH = 85\%)$ or $\gamma(532nm)$, and, very interestingly, a natural classification of the dominating aerosol regime, underlined

by a color code in Figure 6, appears. The highest values of $f_{ref}^{532}(RH = 85\%)$ (> 3) all represent situations with a notable

fraction of sea salt; values of $f_{ref}^{532}(RH = 85\%)$ between 2 and 3 are representative of polluted situations with different

mixings; near the value of 2 we find biomass burning; between 1.5 and 2 rural background with automobile traffic; and values

of $f_{ref}^{532}(RH = 85\%)$ close to 1 correspond to clean and mineral dust cases. Despite the small statistics of this analysis, the fact

that these measurements made in different places of the Earth and in different aerosol loads lead to a coherent classification

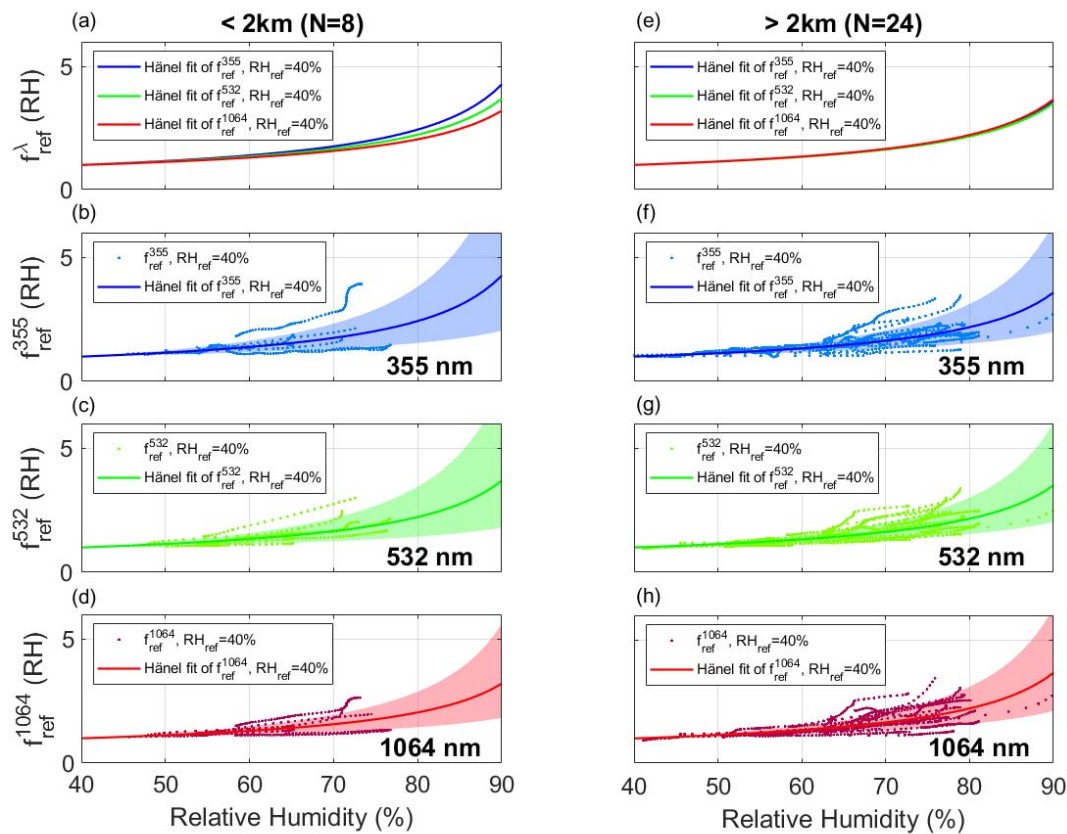


**Figure 5** Particle backscatter coefficient enhancement factors with $RH_{ref}$ = 40 % at 355, 532, and 1064 nm. (a) Spectral Hänel fits for the cases below 2 km at 355, 532, and 1064 nm; (b) Individual $f_{ref}^{355}$ and mean Hänel fit; (c) Individual $f_{ref}^{532}$ and mean Hänel fit; (d) Individual $f_{ref}^{1064}$ and mean Hänel fit; (e-h) same as (a-d) for the cases above 2 km. The shaded areas around the mean Hänel fits represent the standard deviation of all individual fits.





when put all together gives a certain credit to the analysis. It is also corroborated by a certain number of works. Haarig et al. (2017) precisely evaluated the hygroscopic growth of pure sea salt in Barbados and established the top limit presented here ($f_{ref}^{532}(RH = 85\%) = 8.88$). Chen et al. (2019) established that the aerosol hygroscopicity in large cities (pollution) is driven by the hygroscopicity of inorganics emitted essentially by the vehicular transport (nitrates, sulfates and ammonium) and water soluble organic carbonaceous particles. Based on Liu et al. (2014), the following components have a decreasing

hygroscopicity: sulfate acid is the most hygroscopic, then come nitrates, sulfates and water soluble organics. The hygroscopicity ($\gamma$) of smoke particles dominated by carbonaceous organic material from biomass burning has been evaluated by Gomez et al. (2018) to be not larger than 0.4 (equivalent to $f_{ref}^{532}(RH = 85\%) < 1.74$). Finally, relatively clean aerosols and mineral dust have been estimated to be poorly hygroscopic by Chen et al. (2019) and Navas-Guzmán et al. (2019), respectively. All these results from independent studies are consistent with the classification presented in this paper. The

singular spectral behaviour of $f_{ref}^{\lambda}(RH = 85\%)$ or $\gamma(\lambda)$ observed in our study below 2 km was reported, although with smaller values, by Navas-Guzmán et al. (2019) between 355 and 1064 nm for a mixture of biomass burning and rural aerosols and by Pérez-Ramírez et al. (2021) between 355, 532 and 1064 nm for sulfates and organics. Given the little literature on the subject, it is difficult at this point to attribute the decrease of $f_{ref}^{\lambda}(RH = 85\%)$ with increasing wavelength to one type of aerosols or another. However, the quantitative difference observed between our study ($2.68 < f_{ref}^{\lambda}(RH = 85\%) < 3.60$) and Navas-

Guzmán et al. (2019) and Pérez-Ramírez et al. (2021) ($1.50 < f_{ref}^{\lambda}(RH = 85\%) < 1.95$) is most probably due to the presence of marine aerosols in our study which are present neither in Switzerland (Navas-Guzmán et al., 2019) nor in Baltimore-

**Table 1** Spectral enhancement factor $f_{ref}^{\lambda}$ at $RH = 85\%$ and $RH_{ref} = 40\%$. Mean and standard deviation ($\sigma$) of $\gamma(\lambda)$ and the correlation coefficient, $R^2$. Results are given separately for hygroscopic layers below and above 2 km. The layer-mean

Ångström exponent between the pairs (355, 532 nm) and (532, 1064 nm) is reported in the last two columns.

| Centre of the hygroscopic layer (Aerosol type) | Number of cases | Wavelength | $f_{ref}^{\lambda}(RH = 85\%)$ $\pm\,\sigma$ | $\gamma(\lambda)$ $\pm\,\sigma$ | $R^2$ $\pm\,\sigma$ | $(\lambda_1, \lambda_2)$ | $AE_{\lambda_1,\lambda_2}$ |
|---|---|---|---|---|---|---|---|
| < 2 km (Sea salt / Local pollution) | 8 | 355 nm | 3.60 ± 2.47 | 0.81 ± 0.41 | 0.92 ± 0.06 | 355/532 | 1.07 |
| | | 532 nm | 3.18 ± 2.07 | 0.73 ± 0.40 | 0.96 ± 0.04 | 532/1064 | 0.93 |
| | | 1064 nm | 2.68 ± 1.20 | 0.65 ± 0.31 | 0.95 ± 0.04 | | |
| > 2 km (Sea salt / Regional pollution) | 24 | 355 nm | 2.96 ± 1.38 | 0.71 ± 0.32 | 0.91 ± 0.14 | 355/532 | 1.59 |
| | | 532 nm | 2.88 ± 1.27 | 0.70 ± 0.30 | 0.95 ± 0.06 | 532/1064 | 1.28 |
| | | 1064 nm | 2.99 ± 1.38 | 0.73 ± 0.30 | 0.93 ± 0.06 | | |



**Table 2** $f_{ref}^{\lambda}$ at $RH$ = 85% and $RH_{ref}$ = 40 % and $\gamma(\lambda)$ from the literature and from this study (in bold font). Values are listed in a decreasing order of the values at $\lambda$ = 532 nm. SS stands for sea salt, MD for mineral dust and BB for biomass burning. N is the number of cases. Organics[++] indicates a greater amount of organics compared to Organics[+].

| Works | N | Aerosol type | $f_{ref}^{\lambda}(RH = 85\%)$ | | | $\gamma(\lambda)$ | | |
|---|---|---|---|---|---|---|---|---|
| | | | Wavelength (nm) | | | | | |
| | | | 355 | 532 | 1064 | 355 | 532 | 1064 |
| Haarig et al., 2017 | 1 | SS (pure) | 4.10 | 8.88 | 15.95 | 1.08 | 1.49 | 1.84 |
| Granados-Muñoz et al., 2015 | 1 | SS / Sulfates | - | 4.60 | - | - | 1.1 | - |
| Fernandez et al., 2015 | 1 | SS / Nitrates / Organics | - | 3.41 | - | - | 0.88 | - |
| **This study, < 2 km** | **8** | **Local pollution / SS** | **3.60** | **3.18** | **2.68** | **0.81** | **0.73** | **0.65** |
| **This study, > 2 km** | **24** | **Regional pollution / SS** | **2.96** | **2.88** | **2.99** | **0.71** | **0.70** | **0.73** |
| Chen et al., 2019 | 1 | Sulfates / Nitrates / Organics | - | 2.46 | - | - | 0.65 | - |
| Fernandez et al., 2015 | 1 | Nitrates / Organics | - | 2.26 | - | - | 0.59 | - |
| Granados-Muñoz et al., 2015 | 1 | Regional pollution / MD | - | 2.17 | - | - | 0.56 | - |
| Bedoya-Velázquez et al., 2018 | 1 | BB / Regional pollution | 1.74 | 1.95 | - | 0.40 | 0.48 | - |
| Navas-Guzmán et al., 2019 | 1 | BB / Rural | 1.95 | - | 1.50 | 0.48 | - | 0.29 |
| Pérez-Ramirez et al., 2021 | 1 | Sulfates / Organics[++] | 1.89 | 1.72 | 1.54 | 0.46 | 0.39 | 0.31 |
| Pérez-Ramirez et al., 2021 | 1 | Sulfates / Organics[+] | 2.46 | 1.69 | 1.67 | 0.65 | 0.38 | 0.37 |
| Chen et al., 2019 | 1 | Sulfates / Organics (clean) | - | 1.15 | - | - | 0.10 | - |
| Navas-Guzmán et al., 2019 | 1 | MD (nearly pure) | 1.28 | - | 1.18 | 0.18 | - | 0.12 |

Washington (Pérez-Ramírez et al., 2021). The presence of marine aerosols in Barcelona also explains why higher $f_{ref}^{\lambda}(RH = 85\%)$ are found compared to the rest of studies also dominated by pollution (Fernández et al., 2015; Granados-Muñoz et al., 2015; Chen et al., 2019) in which the presence of marine aerosols is not mentioned.

Above 2 km, the particle backscatter coefficient enhancement factor seems to have no spectral dependency: $f_{ref}^{\lambda}(RH = 85\%)$ ($\gamma(\lambda)$) is 2.96 ± 1.38 (0.71 ± 0.32), 2.88 ± 1.27 (0.70 ± 0.30) and 2.99 ± 1.38 (0.73 ± 0.30) at 355, 532 and 1064 nm,

respectively. Such a flat spectral dependency has been observed only by (Navas-Guzmán et al., 2019) between 355 and 1064 nm for mineral dust. We believe that the spectral dependency of the total aerosol content is function of the chemical composition and of the concentration of each one of the hygroscopic components, and for this reason it is highly variable. Although the hygroscopic parameter of relevant particle components has been reviewed by Liu et al. (2014), our conclusion calls for further chemical analysis and laboratory studies to determine the spectral behaviour of these relevant particles.


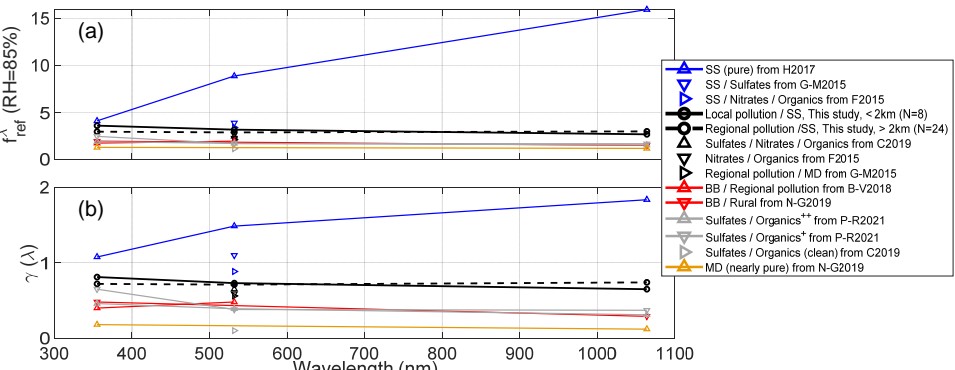

**Figure 6** (a) $f_{ref}^{\lambda}$ at $RH$ = 85% and $RH_{ref}$ = 40 % and (b) $\gamma(\lambda)$ from the literature and from this study as a function of wavelength. Coloured symbols represent the dominating regime: blue for sea salt, black for pollution, red for biomass burning, grey for traffic+rural and clean, and orange for mineral dust. The references are Haarig et al., 2017 (H2017), Granados-Muñoz et al., 2015 (M-G2015), Fernández et al., 2015 (F2015), Chen et al., 2019 (C2019), Bedoya-Velásquez et al., 2018 (B-V2018), Navas-Guzmán et al., 2019 (N-G2019) and Pérez-Ramírez et al., 2021 (P-R2021). Organics$^{++}$ indicates a greater amount of organics compared to Organics$^{+}$.

### 3.2 Climatological analysis at 532 nm

In this section we present for the first time a climatological analysis of the aerosol hygroscopic growth observed along a vertical range in ambient conditions by means of the hygroscopic growth parameter, $\gamma(\lambda)$, and the particle backscatter coefficient enhancement factor at $RH$ = 85 %, $f_{ref}^{\lambda}(RH = 85\%)$. Data from both the multi-wavelength ACTRIS/EARLINET lidar (period 2010-2018) and the MPL (period 2015-2018) are considered. The common wavelength is $\lambda$ = 532 nm. All data were screened according to the selection criteria mentioned in Section 2.2. Like in Section 3.1 all cases associated with a mineral dust intrusion are not included in this study. We find a total of 76 cases distributed along all months of the year. The monthly variation of $\gamma(532)$ is represented in Figure 7. The mean-layer height ($MLH$) and mean-layer relative humidity ($MLRH$) calculated as the mean value in the hygroscopic layer of the height and relative humidity, respectively, is also plotted. The seasonal means of $\gamma(532)$, $MLH$ and $MLRH$ are reported in Table 3. Winter includes the months of December, January and February; spring: March, April and May; summer: June, July and August; and autumn: September, October and November.

From the top plot of Figure 7, one sees that the aerosol hygroscopic growth parameter is in average rather constant all year round. However, for each single month, large standard deviations of $\gamma(532)$ are observed (red shaded area in Figure 7), which indicates the presence of aerosols with different hygroscopic properties all year round. The annual mean of $\gamma(532)$ is 0.55. While the seasonal deviations from that mean are small ($\leq \pm 0.03$, Table 3), the standard deviations associated to each season are large (they vary between $\pm 0.21$ and $\pm 0.25$). The annual mean of the enhancement factor at $RH$ = 85 %, $f_{ref}^{532}(RH = 85\%)$, is $2.26 \pm 0.72$ and the seasonal deviations from that mean are not larger than $\pm 0.08$. Unlike $\gamma(532)$, the $MLH$ shows an annual



cycle: the hygroscopic layers are detected at the highest height in summer (summer $MLH = 2.40$ km) and at the lowest height in winter (winter $MLH = 1.19$ km). In regard of former works of (Sicard et al., 2006) establishing that the planetary boundary layer (PBL) in Barcelona is not significantly different between winter and summer seasons and that it is usually lower than 1.0 km, our findings suggest that hygroscopic layers are detected near the top or slightly above the PBL in autumn and winter, and clearly above the PBL in spring and summer. Although the hygroscopic aerosols are detected above the PBL in spring and

summer, they might not be that different from the aerosols in the PBL. Indeed Pérez et al. (2004) showed that in Barcelona the combined effects of strong insolation, weak synoptic forcing, sea breezes and mountain-induced winds create re-circulations of pollutants injected at various heights above the PBL and up to 4.0 km. Like $\gamma(532)$, the $MLRH$ is also rather constant all year round which confirms that aerosol hygroscopic properties are not related to the level of humidity in the atmosphere. The annual mean, ~70 %, is close to the annual average (72%) at ground level (Weather and Climate, 2021), which is higher than

most of other Spanish cities because of the presence of the sea in Barcelona. The comparison with the literature is not straightforward because of the lack of studies of long-term datasets. Sheridan et al. (2001) and Jefferson et al. (2017) analysed, respectively 1 and 7 years of aerosol growth factors retrieved from scattering coefficients at 550 nm measured with humidified nephelometers in an agricultural region of Southern Great Plains in the United States, a completely different site from Barcelona in terms of aerosol composition (aged aerosols of mostly organic composition vs. anthropogenic, crustal and marine

aerosols). Even if the findings of Jefferson et al. (2017) are based on scattering-derived (and not backscattering) hygroscopic

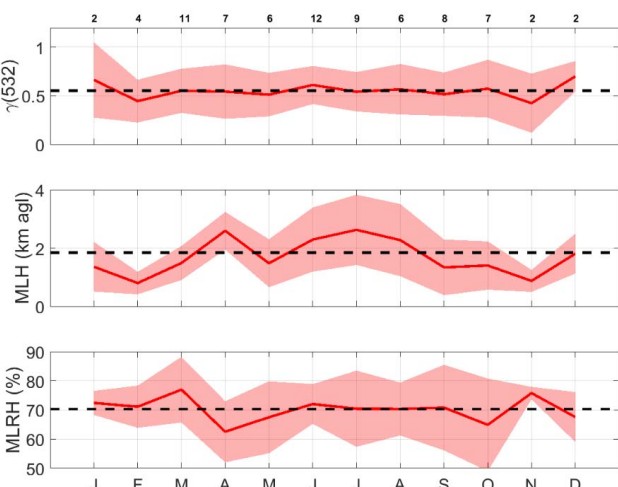

**Figure 7** Monthly (top) hygroscopic growth parameter at 532 nm, $\boldsymbol{\gamma(532)}$; (center) mean-layer height ($\boldsymbol{MLH}$); (bottom) mean-layer relative humidity ($\boldsymbol{MLRH}$). The red shaded area represents the standard deviation. The yearly average is represented with a black dash line. The numbers above the top plot are the number of hygroscopic cases per month.



growth parameters, some differences and similarities with our study are worth mentioning:

- the annual mean of $\gamma$ is 0.40, much lower than in our study (0.55) where the effect of sea salt is noticeable.
- the annual standard deviation of $\gamma$ is 0.15, proportionally similar to $\sigma$ = 0.23 found in our study.
- $\gamma$ is higher in winter (higher nitrate mass fraction) and lower in summer (higher organic mass fraction), whereas it is not season-dependent in our study.

An interesting result from Jefferson et al. (2017), complimentary to our analysis, is the retrieval of $\gamma$ for sub-1 and sub-10 μm particles which annual mean is 0.44 and 0.40, respectively. The lower sub-10 μm values of $\gamma$ is attributed to the influence from soil dust.

**Table 3** Seasonal and yearly mean and standard deviation ($\sigma$) of $f_{ref}^{532}$ at $RH$ = 85% and $RH_{ref}$ = 40 %, $\gamma(532)$, the mean
layer height ($MLH$) and the mean-layer relative humidity ($MLRH$). $N$ is the number of hygroscopic cases. The yearly means are also given for the cases *No cloud* and *Below-cloud*.

| Cases | N | Season | $f_{ref}^{532}(RH = 85\ \%) \pm \sigma$ | $\gamma(532) \pm \sigma$ | $MLH \pm \sigma$ (km) | $MLRH \pm \sigma$ (%) |
|---|---|---|---|---|---|---|
| **All** | **8** | **Winter** | 2.30±0.81 | 0.56±0.25 | 1.19±0.66 | 70.6±6.3 |
| **All** | **24** | **Spring** | 2.22±0.74 | 0.54±0.23 | 1.81±0.82 | 70.4±12.6 |
| **All** | **27** | **Summer** | 2.32±0.70 | 0.58±0.21 | 2.40±1.13 | 71.1±9.4 |
| **All** | **17** | **Autumn** | 2.20±0.77 | 0.53±0.25 | 1.31±0.83 | 69.0±14.3 |
| **All** | **76** | **Year** | 2.26±0.72 | 0.55±0.23 | 1.84±1.03 | 70.3±11.3 |
| *No cloud* | **55** | **Year** | 2.34±0.72 | 0.58±0.22 | 2.12±0.97 | 67.7±11.1 |
| *Below-cloud* | **21** | **Year** | 2.05±0.72 | 0.48±0.24 | 1.11±0.81 | 77.2±8.7 |

Our search for a special dependency of the hygroscopic growth with other factors (layer height, day/night, level of humidity, etc.) was rather unfruitful. However, by looking individually to all 76 cases, and in particular to the vertical profiles of the
backscatter coefficient and relative humidity, we found a possible sub-categorization into 2 classes: cases with no cloud in the vertical range examined (referred hereafter as *No cloud*) and cases where the hygroscopic behaviour was detected just below a cloud (referred hereafter as *Below-cloud*). From the whole dataset 55 cases were classified as *No cloud* and 21 as *Below-cloud*. We have checked that all cases were in sub-saturation humidity conditions ($RH$ < 100 %, the water is in vapour form and the aerosol cannot activate (yet) into a droplet). Figure 8 shows an example of both cases. The annual means of $\gamma(532)$,
$f_{ref}^{532}(RH = 85\%)$, $MLH$ and $MLRH$ for both cases are also reported in the bottom part of Table 3. On a yearly basis the *Below-cloud* cases have a lower $\gamma(532)$ (0.48 vs. 0.58 for the *No cloud* cases) which occur at lower altitude ($MLH$ = 1.11 km vs. 2.12 km) and at higher relative humidity ($MLRH$ = 77.2 % vs. 67.7 %). The *Below-cloud* cases occur inside or near the top of the



PBL where the formation of convective non-precipitating PBL clouds is frequent in coastal sites (Papayannis et al., 2017). In such cases, the relative humidity is higher than in the *No cloud* cases. The aerosol below the cloud starts to activate as cloud

condensation nuclei, its size grows through adsorption of water vapour, its scattering properties increase, and as the aerosol size grows, its potential to keep growing are reduced compared to a drier aerosol. All this is well illustrated in the two cases shown in Figure 8: while both the backscatter coefficient and the relative humidity are low ($RH < 55$ %, $\beta^{532} < 1$ Mm$^{-1}$sr$^{-1}$) in the absence of clouds (Figure 8a), they are high ($RH > 80$ %, $\beta^{532} > 2$ Mm$^{-1}$sr$^{-1}$) and increase strongly with height in the *Below-cloud* case (Figure 8b). So, although one would be tempted to visually attribute the strongest growth to the *Below-cloud*

case, in practice the contrary occurs: $\gamma(532)$ is higher for *No cloud* (0.87) than for *Below-cloud* (0.53). The same result is reflected in the climatological data (Table 3): $\gamma(532) = 0.58$ for *No cloud* and 0.48 for *Below-cloud*. There are at least three reasons for that:

- The first reason has been given above: the aerosol activation as cloud condensation nuclei in high humidity conditions reduces its potential to keep growing compared to the aerosol in drier conditions.

- The second one is mathematical, definition-dependent and inherent to the aerosol composition. According to the definition of the enhancement factor (Eq. 3), defined as a power law function normalized to a $RH_{ref}$ of 40 %, one sees that (bottom plots of Figure 8): 1) in cases with low $RH$, $f_{ref}^{532}$ varies very little and a relatively high $\gamma(532)$ is required to provoke departure of $f_{ref}^{532}$ from unity; and 2) for high $RH$ cases, $f_{ref}^{532}$ varies steeply and a relatively low $\gamma(532)$ is enough to provoke strong variations of $f_{ref}^{532}$.

- The third explanation is linked to the specific aerosol composition at the site. In Figure 9 we show box and whisker plots of the percentiles of the seasonal values of $\gamma(532)$ for all cases, and for the *No cloud* and *Below-cloud* cases. When considering all cases, the results for the monthly statistics is reproduced for the seasonal one: $\gamma(532)$ is not season-dependent. In spring and summer, $\gamma(532)$ for *No cloud* is not significantly different from for all cases; for *Below-cloud* $\gamma(532)$ is slightly larger but well within the seasonal standard deviation (see Table 3). The most

important difference is in autumn when the mean $\gamma(532)$ is 0.71 for *No cloud* and 0.33 for *Below-cloud*, i.e. respectively well above and below the autumn mean of all cases (0.53).

This last paragraph presents a discussion on the possible explanations of the difference observed in autumn in Figure 9 which may rely on the concomitance of several factors. As far as the *No cloud* cases are concerned, in Barcelona hydrophilic inorganics such as nitrates (the most abundant) and ammonium are maximum in autumn and winter, while sulfates are

minimum (Querol et al., 2001). The persistence of anticyclonic stagnating conditions during these seasons favours the accumulation of the pollutants in the PBL (Querol et al., 2001; Pey et al., 2010). In such conditions more pollutants are prone to convective vertical motion and activation as cloud condensation nuclei if high humidity conditions are also present. Gunthe et al. (2011) showed that aged pollution particles in stagnant air (soluble inorganics dominate the mass fraction) are on average larger and more hygroscopic than fresh pollution particles (organics and elemental carbon dominate the mass fraction). Taken

all together, these results support the higher values of $\gamma(532)$ for the *No cloud* cases found in autumn (0.71) compared to the



rest of the year (0.52 – 0.60). Two things in Figure 9 remain to be elucidated: why the hygroscopicity of the *Below-cloud* cases is lower in autumn than during the rest of the year, and why is it lower than that of the *No cloud* cases during the same season? These questions are difficult to answer without complementary in-situ measurements, and at this point, only hypothesis can be formulated. The seasonal statistics available indicates that the mean-layer backscatter coefficient (not shown) in the

hygroscopic layer of the *Below-cloud* cases are larger in autumn (1.71 Mm$^{-1}$sr$^{-1}$) than in spring (0.73 Mm$^{-1}$sr$^{-1}$) and summer (1.35 Mm$^{-1}$sr$^{-1}$); the same occurs for the *MLRH* but in a lesser extent (79.7 % in autumn vs. 74.3 and 76.5 % in spring and summer, respectively); and the opposite for the *MLH* (0.60 km –i.e. within the PBL, see Sicard et al. (2006)– in autumn vs. 1.29 and 1.64 km –i.e. above the PBL– in spring and summer, respectively). Thus, in autumn the *Below-cloud* hygroscopic layers are within the PBL, hence the larger $\beta^{532}$ observed with respect to the spring and summer seasons. The lower autumn

$\gamma(532)$ could reflect a higher fraction of organics in the aerosol mixture in the PBL in autumn vs. a higher amount of inorganics in the aerosol mixture above the PBL in spring and summer (possibly coming from the re-circulation of pollutants injected at various heights above the PBL, see Pérez et al. (2004). Note that this statement is a pure hypothesis. The literature emphasizes the complexity of the atmospheric aerosol hygroscopicity linked to their highly variable composition and chemical transformation. Cheung et al. (2020) suggest that the uptake of hydrophilic/hydrophobic species during particle growth and

coagulation processes may influence the hygroscopicity of aerosols. Cruz and Pandis (2000) study the effect of organic mixing and coating on the hygroscopic behaviour of inorganics and on NaCl particles in particular. Interestingly, they find that, depending on the organic mass fraction, the NaCl-organic mixtures could not only decrease (down to 40%), but also increase (up to 20%) the mixture hygroscopicity. More recently Ruehl and Wilson (2014) emphasize the new and complex relationship between the composition of an organic aerosol and its hygroscopicity and in the same field (Liu et al., 2018) study in the

laboratory some of microphysical mechanisms involved in the hygroscopicity of secondary organic material. All studies call for further laboratory and field research. The difference between the autumn *Below-cloud* (0.33) $\gamma(532)$ and *No cloud* (0.71) is significant. The mean-layer backscatter coefficient in the hygroscopic layer of the *Below-cloud* (1.71 Mm$^{-1}$sr$^{-1}$) is approximately twice larger than the *No cloud* cases (0.80 Mm$^{-1}$sr$^{-1}$). The autumn *MLRH* / *MLH* are higher / lower for the *Below-cloud* (79.7 % / 0.60 km) than for the *No cloud* cases (59.5 % / 1.93 km). It is possible here again to hypothesize that

the lower $\gamma(532)$ for *Below-cloud* could reflect a higher fraction of organics in the aerosol mixture in the PBL with higher humidity conditions vs. a higher amount of inorganics in the aerosol mixture in the free troposphere in the *No cloud* cases. Note that this statement is again a pure hypothesis. In summary, the observations in autumn show that the *Below-cloud* aerosols are detected in the PBL, at high relative humidities, and have large backscatter coefficients and low hygroscopic growth parameters. We close this section with a question: may the activation into CCN at the base of the cloud be affecting

predominantly inorganics salts and thus generating a depletion of them and leave room to an organic-rich layer below the cloud?





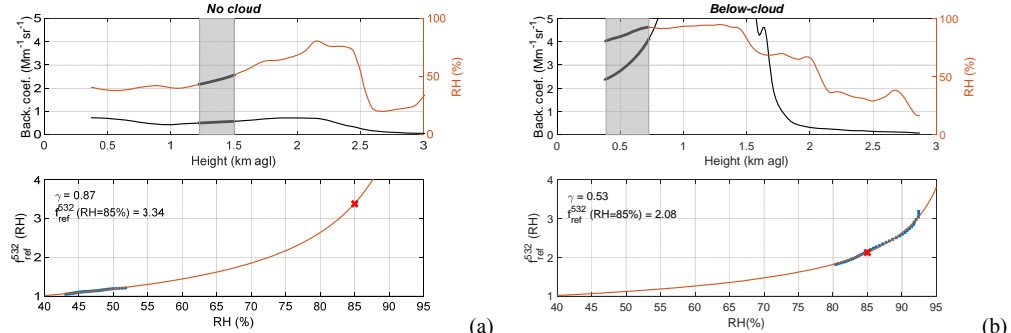

**Figure 8** (a) *No cloud* and (b) *Below-cloud* cases; (top) backscatter coefficient (black, left axis) and relative humidity (red, right axis) versus height; (bottom) $f_{ref}^{532}$ at $RH = 85\%$ and $RH_{ref} = 40\%$ (blue crosses) and Hänel fit (red line). In the top plots 420 the hygroscopic layer is reported in a shaded rectangle and thicker lines.

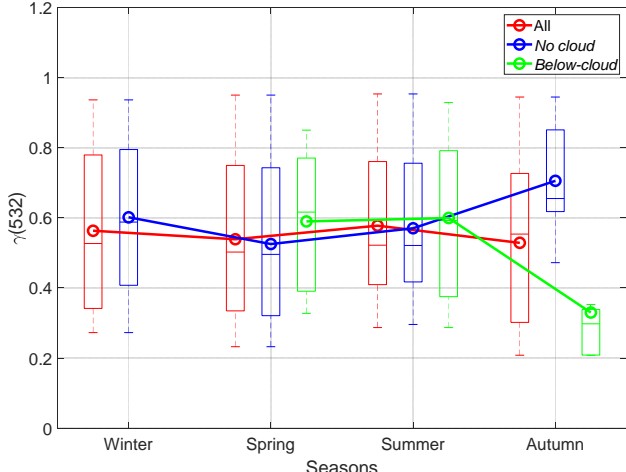

**Figure 9** Box and whisker plot showing the 5, 25, 50, 75, and 95th percentiles of the seasonal values of $\gamma(532)$ for all cases (red), and the *No cloud* (blue) and *Below-cloud* cases (green). Circles indicate the arithmetic mean. Seasonal percentiles and 425 means are shown only when more than one data point is available.





## 4 Conclusions

A spectral and climatological analysis of the aerosol hygrowscopic growth parameter obevered in the atmospheric vertical column combining lidar and radiosoundings measurements is presented. The hygroscopic cases have been selected by filtering time coincident lidar and radiosoundings measurements, by detecting coincident backscatter coefficient and relative humidity

increases with increasing height, and by limiting the variations of variables used as indicators of well mixed conditions such as water vapor mixing ratio, potential temperature, and wind speed and direction. Results are presented in terms of the hygrowscopic growth parameter, $\gamma$, result of fitting the particle backscatter coefficient enhancement factor with the Hänel parametrization and the particle backscatter coefficient enhancement factor, $f_{ref}(RH = 85\%)$, at $RH$=85 % with $RH_{ref}$=40 %. For our results to be comparable with the literature giving enhancement factors for a large variety of $RH_{ref}$ and $RH_{max}$, a

very simple conversion for any values of $RH_{ref}/RH_{max}$ is proposed and applied to the literature values to get $f_{ref}(RH = 85\%)$ with $RH_{ref}$=40 %.

The spectral analysis performed at the wavelengths of 355, 532 and 1064 nm distinguishes aerosols in layers below 2 km (regime of local pollution and sea salt) and above 2 km (regime of regional pollution and residual sea salt). Below 2 km, $\gamma$ decreases with increasing wavelengths ($\gamma$=0.81, 0.73 and 0.65; $f_{ref}^{\lambda}(RH = 85\%)$=3.60, 3.18 and 2.68). This behaviour could

be attributed to the aerosol size: the smaller the aerosol, the more hygroscopic. This hypothesis is supported by the Ångström exponents which are higher for the pair (355, 532) than for (532, 1064), which points out to a spectral sensitivity of the backscatter coefficient slightly larger at shorter wavelengths than at larger wavelengths. Above 2 km the values of $\gamma$ (0.71, 0.70 and 0.73; $f_{ref}^{\lambda}(RH = 85\%)$=2.96, 2.88 and 2.99) are comparable to those below 2 km, and their spectral behaviour is flat. This analysis and others from the literature are put together in a table presenting for the first time spectrally the hygroscopic

growth parameter and enhancement factors of a large variety of atmospheric aerosol hygroscopicities going from low (pure mineral dust, $\gamma < 0.2$; $f_{ref}^{\lambda}(RH = 85\%) < 1.3$) to high (pure sea salt, $\gamma > 1.0$; $f_{ref}^{\lambda}(RH = 85\%) > 4.0$). In this table, the highest values of $f_{ref}^{532}(RH = 85\%)$ (> 3) all represent situations with a notable fraction of sea salt; values of $f_{ref}^{532}(RH = 85\%)$ between 2 and 3 are representative of polluted situations with different mixings; near the value of 2 we find biomass burning; between 1.5 and 2 rural background with automobile traffic; and values of $f_{ref}^{532}(RH = 85\%)$ close to 1 correspond to clean

and mineral dust cases.

The climatological analysis shows that $\gamma$ at 532 nm is rather constant all year round and has a large monthly standard deviation suggesting the presence of aerosols with different hygroscopic properties all year round. The annual $\gamma$ is 0.55±0.23 ($f_{ref}^{532}(RH = 85\%)$=2.26±0.72). The height of the hygroscopic layers shows an annual cycle with a maximum clearly above the PBL in summer and a minimum near the top of the PBL in winter. Although the hygroscopic aerosols are detected above

the PBL in spring and summer, they might not be that different from the aerosols in the PBL. Former works describing the presence of re-circulation layers of pollutants injected at various heights above the PBL may explain why $\gamma$, unlike the height of the hygroscopic layers, is not season-dependent. The sub-categorization of the whole database into *No cloud* and *Below-*





*cloud* cases reveals a large difference of $\gamma$ in autumn between both categories (0.71 and 0.33, respectively), possibly attributed to a depletion of inorganics at the point of activation into cloud condensation nuclei in the *Below-cloud* cases. Our work calls

for more in-situ measurements to synergetically complete studies, like this one, based mostly on remote sensing measurements.

**Data Availability.** The data from the ACTRIS/EARLINET system from the period 2000-2015 can be found in The EARLINET publishing group 2000-2015 et al. (2018), doi:10.1594/WDCC/EARLINET_All_2000-2015. These data are not publicly available following the data policy of CERA (Climate and Environmental Retrieval and Archive) archive database where they are stored. The data of the ACTRIS/EARLINET system from 2016-2018 are freely available at the link

https://actris.nilu.no/. MPL raw data and radiosoundings are freely available but on request.

**Acknowledgments.** The MPLNET project is funded by the NASA Radiation Sciences Program and Earth Observing System. The MPLNET staff at NASA GSFC is warmly acknowledged for the continuous help in keeping the MPL systems and the data analysis up to date.

**Financial support.** This research was funded by the Spanish Ministry of Science and Innovation (PID2019-103886RB-I00),

the Spanish Ministry of Economy, Industry and Competitiveness (CGL2017-90884-REDT), the H2020 programme from the European Union (GA no. 654109, 778349, 871115 and 101008004), and the Units of Excellence "María de Maeztu" (MDM-2016-0600) financed by the Spanish State Research Agency (AEI).

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
