# Peer review of "Measurement Report: Spectral and statistical analysis of aerosol hygroscopic growth from multi-wavelength lidar measurements in Barcelona, Spain"

_Atmospheric Chemistry and Physics, 2021_

## Author Comment (AC1)

This paper deals with a statistical analysis of aerosol higroscopicity determined with the combination of multiwavelength measurements and radiosondes. The topic is of great interest to advance in the water vapor uptake by atmospheric aerosols particles and thus advance in understanding the role of aerosols as potentials cloud condensation nuclei. Of particular interest is the analysis of aerosol hygroscopicity below clouds and for clean skies. Therefore, the study topic is of interest for Atmospheric Chemistry and Physics goals.

To my knowledge, this is one of the first papers that use a long-term database for statistical analyses which implicitly is a novelty. The database of multiwavelength Raman and micropulse lidars plus radiosondes is unique. The paper is well structured and references are updated. Results are unique. The paper has the potential of becoming a reference study for future statistical analyses that involved more ACTRIS stations and similar infrastructures worldwide. However, because of the expectative, I have some concerns that must be addressed before its final publication in Atmospheric Chemistry and Physics.

**Reply**: Thank you very much. We greatly appreciate the reviewer positive feedback.

- Although there are no needs for given specific details about your lidar systems, I would like to know if your Raman lidar measurements used are also for daytime. If so, how does it affect the temporal resolution? Also, it is needed to specify which type of radiosondes used and if data are publicly available.

**Reply**: Until 2020 our multispectral lidar system did not measure Raman signals during daytime. Since 2020 we measure pure Raman rotational signals associated to the elastic wavelength of 355 nm during daytime and make Raman inversions during daytime. Since February 2022 we also measure pure Raman rotational signals associated to the elastic wavelength of 532 nm during daytime. We are currently making tests to check if Raman inversions at 532 nm are feasible during daytime. During daytime, the temporal resolution is degraded compared to nighttime, let's say a factor 3 in general terms. We refer the revierwer to the paper: Zenteno-Hernández, J.A., Comerón, A., Rodríguez-Gómez, A., Muñoz-Porcar, C., D'Amico, G., and Sicard, M.: A Comparative Analysis of Aerosol Optical Coefficients and Their Associated Errors Retrieved from Pure-Rotational and Vibro-Rotational Raman Lidar Signals. Sensors, 21, 1277. https://doi.org/10.3390/s21041277, 2021.

The following sentence has been added line 104: "Radiosoundings are launched automatically by a robotsonde manufactured by Ibatech.". The data are not publicly available but can be obtained on request as it is mentioned in the Data Availability section.

- The methodology proposed for determining aerosol higroscopicity parameters from lidars measurements is not new. The authors give some recent references but I would go the originals papers that explain why the specific thermodynamic conditions must be fulfilled and give a short overview in the paper. But what is not clear to me is the novelty introduced by the authors in using RHref. Moreover, I miss an error analysis in f(RH) and γ because determining their uncertainty is essential for a discussion of similitudes and differences between seasons and between different sites.

**Reply**: With respect to the first point of this comment, the authors propose to replace lines 127-134 by:

For a convective boundary layer, potential temperature, water vapor mixing ratio and wind are conservative variables. Applying thus restrictions to their respective gradients guarantee that the layer is well mixed (Stull, 1976) (Davidson et al., 1984), i.e., the aerosol size distribution is constant with height. In these conditions, changes in the backscatter coefficient can be assumed to be mainly due to uptake of water vapor by particles and not to changes in the aerosol composition or concentration (Feingold & Morley, 2003). Moreover, for all cases, back trajectories at layer maximum and minimum altitudes ($h_{min}$ and $h_{max}$) were also calculated with HYbrid Single-Particle Lagrangian Integrated Trajectory (HYSPLIT) model (Stein et al., 2015) for verifying that all the aerosols inside the layer have the same origin. Such criteria have been applied by other authors (Granados-Muñoz et al., 2015; Navas-Guzmán et al., 2019; among others).

Davidson, K. L., Fairall, C. W., Jones Boyle, P., & Schacher, G. E. (1984). Verification of an Atmospheric Mixed-Layer Model for a Coastal Region. *Journal of Applied Meteorology and Climatology*, *23*(4), 617–636. https://doi.org/10.1175/1520-0450(1984)023<0617:VOAAML>2.0.CO;2

Feingold, G., & Morley, B. (2003). Aerosol hygroscopic properties as measured by lidar and comparison with in situ measurements. *Journal of Geophysical Research: Atmospheres*, *108*(11), 1–11. https://doi.org/10.1029/2002jd002842

Stull, R. B. (1976). The Energetics of Entrainment Across a Density Interface. *Journal of Atmospheric Sciences*, *33*(7), 1260–1267. https://doi.org/10.1175/1520-0469(1976)033<1260:TEOEAD>2.0.CO;2

About the second point of this comment (the novelty introduced in using RHref), we first have removed the adjective "new" to the formulation following one the minor comments of the reviewer (Line 167: I do not understand the statement ' it is the first time a conversion fmin to fref is proposed). Using RHref to re-scale f (from RH=40 to 90 %) is a requisite for being able to compare several enhancement factors obtained for different values of RHmin.

The preparation of the answer to the third and last point of this comment has been the most time-consuming task of this revision. The standard errors associated to both $\gamma$ and $f_{ref}^{\lambda}(RH = 85\%)$ have been calculated and are presented in a new section, "Section 2.3. Calculation of associated standard error". The conclusions of that new section are that the errors found are small enough for the climatological analysis presented in Section 3.2 to be representative of the actual natural atmospheric variations. In order for the reader not to get confused between the standard errors (new Section 2.3) and the standard deviations (Section 3), the following sentence has been added at the beginning of Section 3: "In this section all the standard deviations presented are the actual standard deviation associated to the variable considered and represent the natural variability of the latter.".

- The authors make a great discussion comparing their results with other stations. But they refer many times to chemical properties of aerosols in Barcelona. To me, that should be accompanied with data. I understand that most of your chemical data are from ground-measurements and lidar measurements refer to different altitudes. But

having a study case with well-mixed conditions and chemical properties from the surface would help the discussion.

**Reply**: The chemical information provided in the paper is only coming from references to studies in Barcelona, mostly from Querol et al. (2001) and Pey et al. (2010). We totally agree with the reviewer that tackling the limitation of hygroscopicity characterization with only remote sensing goes through a synergy between insitu (ideally at the ground but also at several levels in the atmosphere) and remote sensing data in an ideally well-mixed atmosphere. The search of such a case study with coinciding insitu chemical analysis and lidar retrievals has not been performed and can, at this stage, only be considered as a future work.

- I get lost with the statistical comparison. The databases for spectral analyses and climatological analyses seem different. That needs a clarification

**Reply**: Only 2 databases are used in this work: the one of the ACTRIS/EARLINET system (2010-2018) and the one of the MPL (2015-2018).

For the spectral analysis only the database of the ACTRIS/EARLINET system is used, together with the strong condition of having simultaneously 3 reliable profiles of backscatter coefficient (at the 3 wavelengths of 355, 532 and 1064 nm).

For the climatological analysis, the database of the ACTRIS/EARLINET system is used and also that of the MPL system. The condition on the database of the ACTRIS/EARLINET system is smoother since only one wavelength (532 nm) is necessary. As a result, the number of cases in the climatological study is higher than in the spectral one.

At the beginning of section 3.1 we have specified the period considered for the ACTRIS/EARLINET system (2010-2018, always the same). In section 3.2 the following sentence in the original manuscript is, in our opinion, clear enough: "Data from both the multi-wavelength ACTRIS/EARLINET lidar (period 2010-2018) and the MPL (period 2015-2018) are considered.".

MINOR COMMENTS

- The time period when data were acquired is not mentioned. That must be clarified

**Reply**: True. This is an omission on our side. The following sentence has been added in the abstract and line 119: "The measurement time considered are those of the radiosoundings, namely 00:00 and 12:00 UTC.".

- Line 46: The statement that aerosol higroscopicity depends on chemical composition need reference. The same for the discussion of deliquescence and cristalization.

**Reply**: The following references have been added for point 1: Orr et al., 1958; and for point 2: Tang et al., 1995; Cziczo et al., 1997; Hansson et al., 1998.

- Line 60: Reference needed for measurements of higroscopicity with tandem nephelometers

**Reply**: The reference of Covert et al. (1972) has been added for the tandem nephelometers, as well as a recent one, Gordon et al. (2015) for spectrometers

- Line 76: I agreed that multispectral lidar measurements are fundamental for aerosol hygroscopic growth study. But this technique has limitations. Please clarify.

**Reply**: The main limitations rely on 1) the properties of interest are retrieved remotely, so that the true values and conditions are never known at 100%, and 2) in case of aerosol mixing the retrieved property is only representative of that mixing which, if only multispectral lidar data are available, cannot be broken down into a detailed aerosol composition. The following sentence has been added line 80: "Aerosol mixing presents a clear limitation of that technique and it is discussed next.".

- Line 97: Details or reference are needed for your algorithm for MPLNET data

**Reply**: A little more information and references have been added about the MPL calibration procedures, starting line 95.

- Lines 140:144: Did you carry out backward-trajectories analyses for stdudy each case?

**Reply**: Yes. This is actually said in the original manuscript: it is in line 131 of the revised one.

- Line 167: I do not understand the statement ' it is the first time a conversion fmin to fref is proposed

**Reply**: The sentence has been deleted.

- Figure 3: Are the data those of Figure 1?

**Reply**: Yes, they are. It is now said in the caption of Figure 1.

- Line 229: I do not understand the expression 'en passant'. Generally I would recommend double-checking English grammar.

**Reply**: "en passant" is a foreign (French) term accepted in the Oxford English dictionary (https://www.oxfordreference.com/view/10.1093/acref/9780199891573.001.0001/acref-9780199891573-e-2161?rskey=nZTKEl&result=1) and Cambridge English dictionary (https://dictionary.cambridge.org/dictionary/english/en-passant) which means 'in passing': if you say something en passant, you mention it quickly while talking about something else.

- Line 277: Do you expect marine aerosols above 2 km? Please clarify

**Reply**: Line 277 is the discussion about the results found below 2 km. The discussion about the results found above 2 km starts line 278 and no mention is made of marine aerosols.

The presence of marine aerosols above 2 km is mentioned at the beginning of Section 3.1. In both references (Sicard et al., 2011 and Pandolfi et al. ,2013) used to define the aerosol category as "Regional pollution and marine aerosols" marine aerosols are mentioned in heights above 2 km. Sicard et al. (2011) emphasize the combined effect of sea breeze and the orography to produce recirculation layers above the PBL, while Pandolfi et al. (2013) evoke the transport of Atlantic air masses. To be a bit clearer, we have added the following explanation in the revised manuscript, line 232: "caused by the sea-breeze phenomenon".

- Line 313: Please clarify the statement 'our findings suggest that hygroscopic layers near the top or slightly above the PBL'.

**Reply**: This sentence has been rephrased as follows: "In regard of former works of Sicard et al. (2006) establishing that the planetary boundary layer (PBL) in Barcelona is not significantly different between winter and summer seasons and that it is usually lower than 1.0 km, our findings suggest that hygroscopic layers in autumn ($MLH$ = 1.31 km) and winter ($MLH$ = 1.19 km) are detected near the top or slightly above the climatological mean PBL height, and clearly above the PBL in spring ($MLH$ = 1.81 km)  and summer ($MLH$ = 2.40 km).".

- Table 3: Why the analyses are only done at 532 nm?

**Reply**: Table 3 is part of Section 3.2 about the climatological analysis performed at 532 nm. Not enough data are available to make a climatological analysis at the 3 wavelengths.

- Lines 355-356: Why as the aerosol size grows, its potential to keep growing are reduced compared to a drier aerosol?

**Reply**: The meaning of this sentence is to express that there is an upper size limit ($D_{lim}$) which marks the transition between a particle and a cloud droplet. The size of an aerosol at 75 % humidity (to give an example), D(75%), will be closer to that size limit than the size of a particle at 40 % humidity, D(40%), because D(40%) < D(75%) < $D_{lim}$. And therefore its potential to keep growing (from D(75%) to $D_{lim}$) is reduced compared to a drier aerosol (from D(40%) to $D_{lim}$). We leave this point up to the reviewer if he/she thinks that after this explanation, the text still needs rewording.

---

## Author Response (AR1)

**Para obtener la mejor experiencia, abra esta cartera PDF en Acrobat X o Adobe Reader X, o en alguna versión posterior.**

¡Consiga Adobe Reader ahora!